DISCOVERY REPORT

# Eggs of the mosquito *Aedes aegypti* survive desiccation by rewiring their polyamine and lipid metabolism

**Anjana Prasad[1,2], Sreesa Sreedharan[2,3], Baskar Bakthavachalu**⦿[1,4]*, **Sunil Laxman**⦿[2]*

**1** Tata Institute for Genetics and Society (TIGS) Centre at inStem, Bangalore, India, **2** Institute for Stem Cell Science and Regenerative Medicine (DBT-inStem), Bangalore, India, **3** SASTRA University, Thirumalaisamudram, Thanjavur, India, **4** School of Biosciences and Bioengineering, Indian Institute of Technology Mandi, Mandi, India

* baskar@iitmandi.ac.in (BB); sunil@instem.res.in (SL)

The Editors encourage authors to publish research updates to this article type. Please follow the link in the citation below to view any related articles.

## Abstract

Upon water loss, some organisms pause their life cycles and escape death. While widespread in microbes, this is less common in animals. *Aedes* mosquitoes are vectors for viral diseases. *Aedes* eggs can survive dry environments, but molecular and cellular principles enabling egg survival through desiccation remain unknown. In this report, we find that *Aedes aegypti* eggs, in contrast to *Anopheles stephensi*, survive desiccation by acquiring desiccation tolerance at a late developmental stage. We uncover unique proteome and metabolic state changes in *Aedes* embryos during desiccation that reflect reduced central carbon metabolism, rewiring towards polyamine production, and enhanced lipid utilisation for energy and polyamine synthesis. Using inhibitors targeting these processes in blood-fed mosquitoes that lay eggs, we infer a two-step process of desiccation tolerance in *Aedes* eggs. The metabolic rewiring towards lipid breakdown and dependent polyamine accumulation confers resistance to desiccation. Furthermore, rapid lipid breakdown is required to fuel energetic requirements upon water reentry to enable larval hatching and survival upon rehydration. This study is fundamental to understanding *Aedes* embryo survival and in controlling the spread of these mosquitoes.

## Introduction

Life as we know has evolved with water. The fundamental unit of life, cells, are made up of water, inorganic ions, and organic compounds. Of these, water makes up a bulk of the cell volume and mass [1,2]. Water is an active constituent of cells both structurally and as a polar, amphoteric reagent [3,4]. The versatility and adaptability of water enable important chemical reactions within a cell, maintaining their structure and function [5]. This makes water vital for life and given this, it is remarkable that some organisms can survive in the near absence of water [6–8]. The phenomenon of survival after desiccation is common in unicellular microbes; is selectively observed in some plants, rotifers, nematodes, larvae of certain insects; and is seldom found in other organisms [9]. The loss of water and the associated volume reduction

**Data Availability Statement:** All relevant data are provided in the manuscript and supplemental information. Mass spectrometry proteomics data are also deposited on the PRIDE database and the data is available via ProteomeXchange with identifier PXD044525. Figure legends indicate the supplemental Tables where raw numerical data are provided.

**Funding:** No specific funding was obtained for this study. DST-INSPIRE (IF190149 to SS) and DBT/Wellcome Trust India Alliance (IA/I/19/1/504286 to BB) supported individual fellowships. These funders had no role in study design, data collection and analysis, support for experiments, decision to publish, or preparation of the manuscript. Intramural support was provided by the Tata Institute for Genetics and Society (to BB), and DBT-inStem (to SL).

**Competing interests:** The authors have declared that no competing interests exist

**Abbreviations:** DFMO, difluoromethylornithine; ETC, electron transport chain; HAE, hours after egg laying; IDP, intrinsically disordered protein; ODC, ornithine decarboxylase; PBS, phosphate-buffered saline; PPP, pentose phosphate pathway; ROS, reactive oxygen species.

leads to the destruction of cell components, severe mechanical stress, DNA and RNA damage, redox imbalances that lead to oxidative stress, protein denaturation, and the formation of toxic aggregates [8,10]. In general, a cell must protect its integrity, preserve protein function, and protect its genome to survive water loss. Our current understanding of cellular and molecular processes that enable desiccation tolerance in some organisms comes from a limited number of model systems. From organisms like yeasts, tardigrades, rotifers, and nematodes, the following desiccation response mechanisms have emerged, which suggest processes by which cellular integrity and function can be preserved. Some organisms accumulate the disaccharide trehalose, which replaces water and prevents protein denaturation and changes in membrane conformation [11–18]. Other organisms induce intrinsically disordered or chaperone proteins that protect other proteins from denaturation, as well as against damage due to oxidative stress [19,20]. Yet other induced processes include polyamine biosynthesis, xenobiotic detoxification, and lipid metabolism [12]. All these suggest diverse processes with convergent functions that help protect a cell as water is lost. The converse processes, of how cells restore function upon rehydration remains more obscure. Understanding mechanisms through which some cells and organisms escape death due to desiccation is therefore of fundamental importance, with implications for agriculture, pest control, and regenerative biology.

In addition to the more studied model organisms, insects including beetles, termites, crickets, and chironomids exhibit desiccation tolerance [21]. The large, sleeping African midge, *P. vanderplanki* can survive desiccation for several years [17]. In this context, mosquitoes are vectors for numerous diseases and are quintessential examples of insects that require water to breed in. With increasing global climate change, mosquitoes have adapted and survive extreme environmental conditions [22–25]. In order to overcome periods of unfavourable conditions, they adopt several biological strategies including suspended development during diapause [26] or more extreme survival strategies which are often characterised by altered metabolism and enhanced tolerance to environmental stressors such as desiccation, oxidative stress, etc. [27]. This is followed by the resumption of growth and reproduction upon restoration of optimal conditions [27]. Such an observation has been made in the eggs of some mosquito species such as *Aedes* that are vectors for several arboviral diseases including dengue, Zika, yellow fever, and Chikungunya [28–31]. Such strategies allow the *Aedes* mosquitoes to globally expand beyond their original habitats in subtropical North Africa and result in frequent outbreaks of arboviral infections [25,27,32,33].

Notably, *Aedes* mosquitoes require water for oviposition and eggs can hatch only in water. However, the eggs, at a late developmental stage, become tolerant to desiccation and can remain desiccated but viable for prolonged periods of time—a year or more. This process is reversible and eggs hatch into larvae upon contact with water again [28–31,34,35]. The mosquito egg is essentially a closed system where all the nutrients required for completing embryogenesis are deposited within the egg before the onset of desiccation, and the eggs do not rely on exogenous supplies except for oxygen and water [36]. While this remarkable ability of *Aedes* eggs to survive despite loss of water is known, any molecular mechanisms associated with desiccation tolerance or survival after rehydration in *Aedes* eggs are unknown. Here, we decipher the nature of biochemical changes in the eggs of *Ae. aegypti* that enable tolerance to extended periods of desiccation, as well as subsequent hatching and revival of eggs after rehydration. We uncover both general, as well as unique biochemical underpinnings in the *Aedes* mosquito eggs that enable desiccation tolerance and revival. These findings provide a biochemical basis for desiccation tolerance in a major vector of arboviral diseases.

## Results and discussion

### *Aedes* eggs acquire desiccation tolerance during late embryonic development

Field studies with *Ae. aegypti* found that, its eggs are resistant to desiccation, survive for months in a dormant state, and hatch into first instar larvae when submerged in water [29]. To systematically understand this process, we developed a controlled, quantifiable desiccation and hatching assay for *Aedes* eggs, as well as the eggs of *Anopheles stephensi*, a desiccation–sensitive mosquito species. Synchronised eggs were collected and allowed to complete embryonation under moist conditions for 48 h (Figs 1A and S1A). A portion of these eggs were subjected to desiccation for a total period of 21 days in separate batches, rehydrated at different time points and the number of larvae that hatched were counted (see Materials and methods, S1A Fig). Eggs from the 0-day batch that were not subjected to desiccation (fresh eggs) were observed under a stereomicroscope, and these eggs appeared intact and healthy. In contrast, eggs subjected to 21 days of desiccation, appeared deformed with inward shrinkage (Fig 1B, top). These eggs had an approximately 65% reduction in total mass (Fig 1B). Notably, the *Aedes* eggs showed excellent viability upon rehydration. When 21 days desiccated *Aedes* eggs were rehydrated, approximately 85% of the eggs rapidly hatched into viable larvae (Fig 1C). The eggs from all the batches tested hatched within 30 min of placing them in water. In contrast, *An. stephensi* eggs did not resist even mild desiccation, and no eggs hatched even after 3 days of desiccation (Fig 1C). This establishes that *Ae. aegypti* eggs can tolerate and survive desiccation. To further ensure that the differences in hatching were not due to variations in the developmental state of the embryos, we clarified fresh and desiccated (21 days) *Aedes* eggs to observe the morphology of the embryo (S1B Fig). Embryos from both fresh and desiccated eggs revealed a well-defined head, visible eyes, thorax, and 10 abdominal segments as previously described for *Aedes* embryogenesis [37]. These data indicate that embryos in both fresh and desiccated eggs were almost completely developed and at closely comparable stages.

Under favourable conditions, *Aedes* eggs require about 48 to 72 h to completely develop and hatch into first instar larvae [31,35,38,39]. In order to determine if eggs have to attain a specific developmental stage to acquire desiccation tolerance, we systematically subjected *Aedes* eggs to desiccation—4, 8, 15, 24, and 48 h post egg laying. Eggs at these different stages of embryonic development were desiccated for a total period of 10 days, following which they were rehydrated (see Materials and methods). We observed that only eggs subjected to desiccation at least 15 h post egg laying remained viable (Fig 1D). In contrast, all eggs with less than 15 h of development before desiccation failed to hatch upon rehydration. We therefore draw 2 conclusions from these data. First, for desiccation tolerance, embryos have to develop until the formation of the serosal cuticle along with the 2 eggshell layers (Fig 1D, top), consistent with earlier studies [34,35,40]. However, this alone is insufficient, since all mosquito eggs, including the eggs of desiccation intolerant *An. stephensi* also has a serosal cuticle. The resistance to desiccation specifically in *Aedes* eggs must therefore come from other factors within these eggs.

### Desiccation does not affect larval development post hatching

The embryos in their fresh and desiccated states exhibited comparable developmental stages as indicated in S1B Fig. However, to investigate potential disparities in the first instar larvae hatching from fresh versus desiccated eggs, we measured larval length everyday post hatching. There was a steady increase in the lengths, without any significant difference between the hatchlings from fresh eggs and those from desiccated eggs (Fig 1E). There was also no difference in the percentage of pupation and eclosure between hatched fresh eggs and desiccated

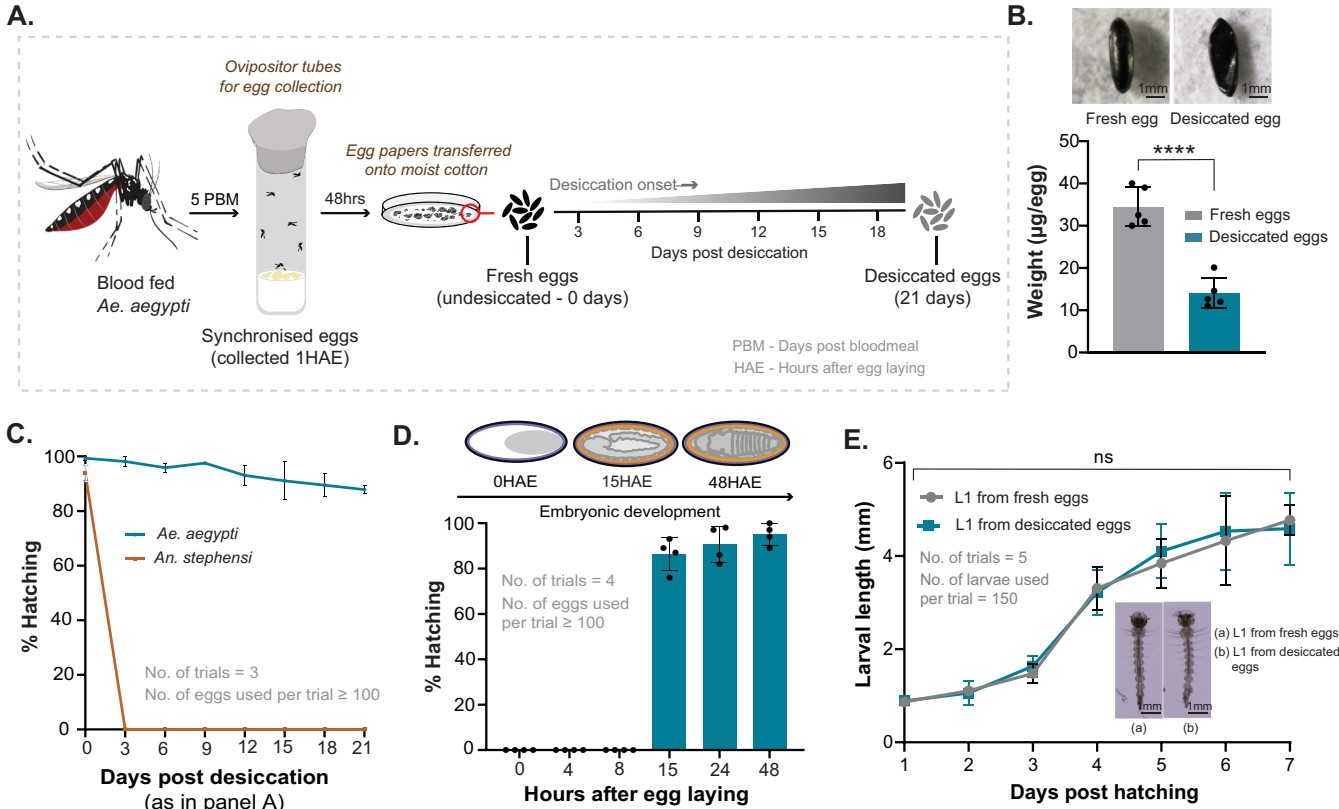

**Fig 1. *Aedes* eggs acquire desiccation tolerance during late embryonic development.** (A) Schematic depicting synchronised *Aedes* egg collection and egg desiccation. Synchronised eggs were collected in ovipositor tubes 5 days post blood meal. Collected eggs were allowed to complete embryonation for 48 h. One batch of eggs stayed hydrated (fresh eggs/0 days post desiccation). Other batches of eggs were placed under moisture for 48 h and subsequently desiccated for 3, 6, 9, 12, 15, 18, and 21 days. See S1A Fig for a detailed schematic of desiccation assay. (B) Morphology and weight changes in desiccated eggs. *Aedes* eggs before (top left) and after desiccation (top right). Desiccated eggs (21 days) appear deformed and shrunk inwards, whereas fresh eggs were oval-shaped and healthy. The graph shows reduction in the weight of desiccated eggs compared to that of fresh eggs. Scale bar = 1,000 μm. The number of trials = 5. Data is represented as mean ± SD. See S1B Fig for the morphology of the embryo within fresh and desiccated eggs. (C) Larval hatching post egg desiccation in *Ae. aegypti* and *An. stephensi* eggs. The graph shows the percentage of desiccated *Aedes* and *Anopheles* eggs hatching over a period of 21 days. Data is represented as mean ± SD. The number of trials = 3, number of eggs used per trial $\geq$ 100. (D) Embryonic development stages in *Aedes* egg resulting in resistance to desiccation. The schematic (top) shows the structure of a mosquito embryo and the egg shell layers over time during embryogenesis (denoted as HAE). Egg shell layers in the schematic are represented as: black—exochorion, blue—endochorion, orange—serosal cuticle, grey dotted lines—serosa, grey—developing embryo. The graph shows percentage of desiccated *Aedes* eggs hatching, when dried at different stages of embryonic development. Data is represented as mean ± SD. The number of trials = 4. The number of eggs used per trial $\geq$ 100. (E) Desiccation and larval development. The growth of first instar larvae hatching from fresh or desiccated eggs was measured in terms of larval length. The number of trials = 5. The number of larvae used per trial = 150. Inset: The morphology of *Aedes* first instar larvae hatching from fresh (left) and desiccated eggs (right). Scale bar = 1,000 μm. Also see S1C and S1D Fig for details on larval development post hatching from either fresh eggs or desiccated eggs into pupa and adult mosquitoes. Statistical significance was calculated using an unpaired Student *t* test. *$p < 0.05$, **$p < 0.01$, ***$p < 0.001$, ns—no significant difference. Raw data underlying Fig 1B, 1C, 1D and 1E can be found in S5 Table.

eggs (S1D Fig). Further, there was no delay in the larval development as seen from the duration of pupation and eclosure in the larvae/pupae emerging from fresh or desiccated eggs (S1C Fig). We also qualitatively examined total protein in first instar larvae after they emerged from fresh or desiccated eggs, after 1 h of hatching. The whole protein extracted from these larvae was resolved and visualised on an SDS-PAGE gel. At a purely qualitative level, these samples from larvae hatching from fresh or desiccated eggs did not show visible differences in their protein profiles (S1E Fig). Collectively, these data indicate that the rehydrated eggs develop into largely normal first instar larvae and that desiccation does not alter larval development post hatching.

## Desiccated *Aedes* eggs remodel their proteome towards lipid metabolism and the TCA cycle

We next asked if during desiccation, the *Aedes* eggs underwent any proteome level changes. For this, we adopted a differential proteome-based approach to identify key proteins that change in desiccated eggs. Whole protein extracted from fresh eggs v/s desiccated eggs (S1F Fig) was resolved on gradient SDS-PAGE gels and stained with Coomassie to visualise proteome-level changes. The stained protein gel revealed obvious differences in protein profiles (Fig 2A) between fresh and desiccated *Aedes* eggs. Some proteins visibly increased in the eggs of *Aedes* post desiccation while some decreased. Protein bands from each lane were excised, proteins extracted and identified using mass spectrometric approaches (see Materials and methods).

We analysed the MS data to identify proteins and pathways that were uniquely induced upon desiccation. A total of 2,141 and 1,837 proteins were identified in fresh *Aedes* eggs (replicate 1 and 2, respectively), and 1,802 and 1,777 proteins were identified in desiccated eggs (replicate 1 and 2, respectively). Out of these, only those proteins with a significant peptide match of more than 2, and with significant emPAI scores [41], were considered for further analysis. The emPAI score of a given protein is proportional to its abundance in the sample [41], based on observed to observable peptides detected by mass spectrometry. We compared the emPAI values of a protein in fresh eggs to that of desiccated eggs, and further grouped the data into increased or decreased proteins. Post desiccation, 45 proteins increased and 125 proteins decreased in amounts. The abundance of 30 proteins did not change during desiccation. Unique protein IDs obtained were used to map functional domains using the *Aedes* genome [42], and relevant biological processes these proteins are involved in were assigned (see S1 Table for a complete list of proteins increasing or decreasing post desiccation).

This analysis revealed that the *Aedes* eggs in the desiccated, stress resistant state differed strikingly from fresh eggs at the protein level. We observed significant enrichment of TCA cycle enzymes as partly increased and partly decreased in desiccated eggs (Figs 2B and S2A). There was also an increased abundance of lipid catabolism enzymes that included lipases and fatty acid oxidation enzymes as shown in Fig 2B and 2C. Furthermore, the desiccated *Aedes* eggs also had increased superoxide dismutase, glutathione transferase, and thioredoxin peroxidase that affect redox balance (Fig 2C). Studies in other organisms suggest that desiccation results in the production of reactive oxygen species (ROS) [12,43,44]. Consistent with these observations, *Aedes* eggs also have increased dismutase, catalase, or peroxidase levels. This might allow desiccated eggs to manage oxidative stress that might occur during desiccation. Additionally, proteins denature (and therefore aggregate or precipitate) due to water loss and countering this requires the activity of protective molecular chaperones [12]. We also observed increased amounts of a few protein chaperones that assist in protein folding (Fig 2C). These components of proteome rewiring related to oxidative stress and protein chaperones in *Aedes* eggs are entirely consistent with observations made in other desiccation tolerant organisms [12,43] and appear to be universal strategies to combat desiccation stress.

Notably, a unique proteome-level rewiring of metabolism could be constructed and we could organise this into a putative, underlying hierarchy. The pathway map (Fig 2B) shows relative changes in central carbon metabolism enzymes from glycolysis, the pentose phosphate pathway (PPP), TCA cycle, and β-oxidation of fatty acids. The enzymes of glycolysis and pentose phosphate decreased (Figs 2B and S2A). In contrast, most enzymes for fatty acid oxidation and only those of the upper arm of TCA cycle (up to α-ketoglutarate) increased (Figs 2B and S2A). This suggested a precise metabolic rewiring that was apparent at the proteome level. Here, the "growth and anabolism"-related metabolic processes of glycolysis and the PPP [45],

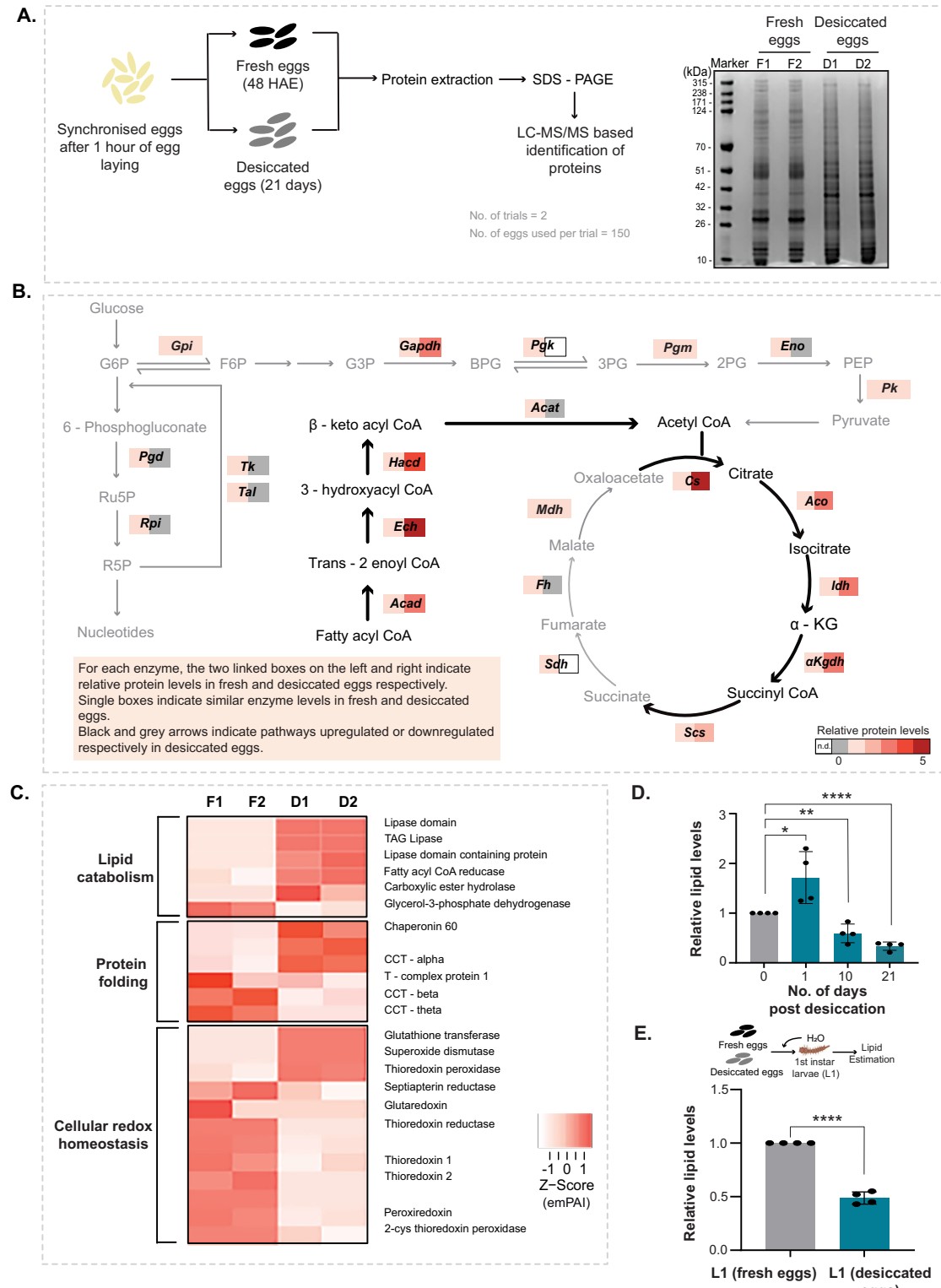

**Fig 2. Desiccated *Aedes* eggs remodel their proteome towards lipid metabolism and TCA cycle.** (A) Proteome changes in *Aedes* eggs post desiccation. (Left) Schematic describing the experimental setup. Also see S1F Fig for an illustration of *Aedes* egg collection to prepare protein extracts. (Right) Coomassie stained SDS-PAGE gel of fresh and desiccated eggs, which show a clear difference in band pattern. F1 and F2 represent 2 independent trials from fresh eggs and D1 and D2 represent 2 trials from desiccated eggs. Number of eggs used per trial = 150. Note: For each lane in the gel 150 eggs (either fresh or desiccated) were pooled and lysed

together for protein extraction. (B) Analysis of proteome changes in fresh and desiccation eggs. The pathway maps of glycolysis, the PPP, the TCA cycle, and the β-oxidation of fatty acids are shown, and the colour-coded boxes show relative changes in protein levels of enzymes in these pathways based on average emPAI scores for that protein (left: fresh eggs, right: desiccated eggs). Note: β-oxidation and the upper arm of the TCA cycle increase in desiccated eggs, while the PPP, glycolysis, and the lower arm of the TCA cycle decrease. Also see S2A Fig for gene ontology-based analysis and grouping of proteins that change during desiccation. (C) The heatmap represents differential protein expression in fresh v/s desiccated eggs, for key proteins from 3 groups: lipid metabolism, protein folding, and redox homeostasis. The colour corresponds to the emPAI score converted to a Z-score for that protein. F1 and F2 and D1 and D2 represent 2 independent trials from fresh eggs and D1 and D2 represent 2 trials from desiccated eggs. Also see S2A Fig for gene ontology-based analysis and grouping of proteins that change during desiccation. (D) *Aedes* eggs and changes in total lipids upon desiccation. The graph represents relative lipid levels in fresh eggs and 1, 10, and 21 days desiccated eggs. Data is represented as mean ± SD. The number of trials = 4. The number of eggs used per trial = 50. Also see S1F Fig which illustrates how *Aedes* eggs were collected for the preparation of lipid extracts. (E) Total lipid levels in first instars hatching from desiccated eggs. Schematic at the top shows the experimental setup. Total lipids were estimated in first instar larvae hatching from fresh and desiccated eggs 1 h post rehydration. The graph represents relative lipid levels in first instar larvae hatching from fresh eggs and those hatching from 21 days desiccated eggs. Data is represented as mean ± SD. The number of trials = 4. The number of larvae used per trial = 50. Statistical significance was calculated using an unpaired Student *t* test. *$p < 0.05$, **$p < 0.01$, ***$p < 0.001$, ns—no significant difference. Raw data and datasets for Fig 2B, 2C, 2D, 2E can be found in S1 and S5 Tables.

as well as the ATP—and NADH—producing arms of the TCA cycle (the post α-ketoglutarate part of the cycle) decreased. In contrast, enzymes of the first 3 steps leading to the α-ketoglutarate arm of the TCA cycle and lipid breakdown were high in desiccated eggs. To address if this was reflected biochemically, we first measured lipid levels in fresh and desiccated *Aedes* eggs (Fig 2D). The desiccated eggs had significantly lower amounts of lipids (Fig 2D). To further resolve this observation, we estimated lipids over time, after eggs were subjected to desiccation. We observed increased lipids in eggs subjected to desiccation stress for 1 day, followed by a gradual decrease (Fig 2D). These data suggest that lipids were synthesised by the eggs immediately after sensing desiccation. As the desiccation phase advances, the accumulated lipids were broken down via fatty acid β-oxidation. We therefore next compared the levels of lipids in first instars hatching from fresh eggs v/s those hatching from 21 days old eggs. Consistently, we observed reduced levels of lipids in larvae hatching from desiccated eggs (Fig 2E). Our data also points towards possible changes in the nature of proteome over the course of desiccation, which may provide interesting insights about the underlying molecular mechanisms in future analysis.

The overall proteome level changes in mosquito eggs therefore reflect what appears to be primarily a metabolic rewiring. In 2 other anhydrobiotes, the dauer stage of the nematode *C. elegans*, and in tardigrades, desiccation increases the amounts of intrinsically disordered proteins or IDPs [12,19,20], and these proteins are thought to provide protection analogous to protein chaperones. In contrast, in desiccated *Aedes* eggs, there is no induction of proteins that might fall into this category of IDPs, nor are there any orthologs of the tardigrade or nematode IDP proteins present in the *Aedes* genome. We therefore minimise the possibility of induced IDPs being a mechanism of desiccation tolerance in mosquito eggs and hypothesize a primarily metabolic rewiring-based acquisition of desiccation tolerance in these eggs.

## Desiccated eggs acquire a hypometabolic state with increased polyamine production

Desiccation tolerance has classically been described as an "ametabolic" phase based on a perceived lack of metabolic activity [46]. However, this is an oversimplification, based primarily on an observed reduction in respiration and energy metabolism (and not all metabolism). A cell undergoing a desiccation/rehydration cycle undergoes several changes during both the transitions to and out of the desiccated state. Following entry into the desiccation phase, cells lose higher-order functions such as motility and cell division leading to developmental arrest [47–49]. With the progression of the desiccation phase, cells up-regulate pathways producing

"stress protectants," conserve energy and retain only basal metabolism for maintenance and repair [17,50–53]. The production of these "stress protectants" require a rewiring of metabolic flux towards their synthesis. It is therefore appropriate to describe desiccation as "hypometabolic," with reduced energy-producing pathways and rewiring of carbon metabolism to support the production of protective molecules.

Based on the distinct proteome rewiring observed during desiccation, we constructed a hypothetical metabolic program in the *Aedes* egg correlated with desiccation tolerance. We asked if this remodelled metabolic state in desiccated *Aedes* eggs retained features of other desiccation tolerant organisms. Desiccation induces oxidative stress and increased generation of ROS that leads to reduced cell viability [44,54]. Studies in multiple organisms find a down-regulation of the TCA cycle [11,17,27,55–57]. This not just reduces ATP production, but will also concurrently reduce the production of NADH and ROS from the electron transport chain (ETC). Our proteomics data suggested a possible decrease in glycolysis as well as the lower arm of the TCA cycle. To directly assess this, we first estimated steady-state levels of key glycolytic and PPP intermediates. The amounts of these intermediates were either reduced or remained constant in desiccated eggs (Figs 3A and S3A). This is consistent with the proteomics data and indicates that these eggs have reduced glycolysis and PPP. We next assessed amounts of the TCA cycle and related metabolites (Fig 3A). Here, we observed a small increase in citrate/isocitrate (in the early part of the TCA cycle), but reduced TCA metabolites from the later part of the cycle. This would be entirely consistent with a reduced (complete) TCA cycle. Taken together, our data indicates a reduction in the key metabolic pathways involved in energy production.

An added outcome of reduced glycolysis can in some cases be an increase in trehalose synthesis, due to rerouted glucose metabolic flux [58,59]. Trehalose is a versatile molecule, and in nematodes and yeasts, carbon flux towards trehalose synthesis increases during desiccation [11,14,15]. To assess if this occurred in *Aedes* eggs, we estimated trehalose amounts in fresh and desiccated eggs (S3B Fig). Notably, in *Aedes* eggs, the amounts of trehalose were very low and did not increase post desiccation (S3B Fig). Note: Controls that included trehalose estimates from comparable biomass of budding yeast grown in glucose in log-phase, which at this stage are themselves not desiccation resistant [11], had an order of magnitude greater amounts of trehalose than the *Aedes* eggs (S3B Fig). These data diminish the possibility of protective roles of trehalose in *Aedes* eggs during desiccation.

An interesting trend observed earlier was that only the upper arm of the TCA cycle remained high (Fig 2B). This could therefore suggest a more nuanced metabolic rewiring. Notably, the step which leads to the production of α-ketoglutarate is the first step towards glutamate, glutamine, arginine, and proline synthesis. Furthermore, arginine is the precursor for polyamine synthesis. We therefore hypothesised that an increased arm of the TCA cycle leading to α-ketoglutarate results in a diversion towards production of these other molecules, while reducing the complete TCA cycle. To test this, we first measured amounts of amino acids in fresh and desiccated eggs of *Aedes*. Notably, the levels of glutamine and arginine increased in desiccated eggs (Figs 3A and S3C). We next measured the levels of polyamines in fresh and desiccated eggs. The polyamines measured—ornithine, putrescine, and spermidine increased substantially in desiccated eggs (Fig 3B). As an added comparison, we also estimated polyamine levels in the desiccation–sensitive *An. stephensi* eggs. Notably, these polyamines decreased in amounts in these eggs (S3D Fig).

Collectively, we find that during the process of desiccation, there is a rewiring of metabolism to a hypometabolic state with reduced glycolysis and the TCA cycle, suggesting reduced energy synthesis. Notably, carbon flux is rerouted away from energy metabolism and towards the production of amino acids, particularly arginine and glutamine. Subsequently, polyamines,

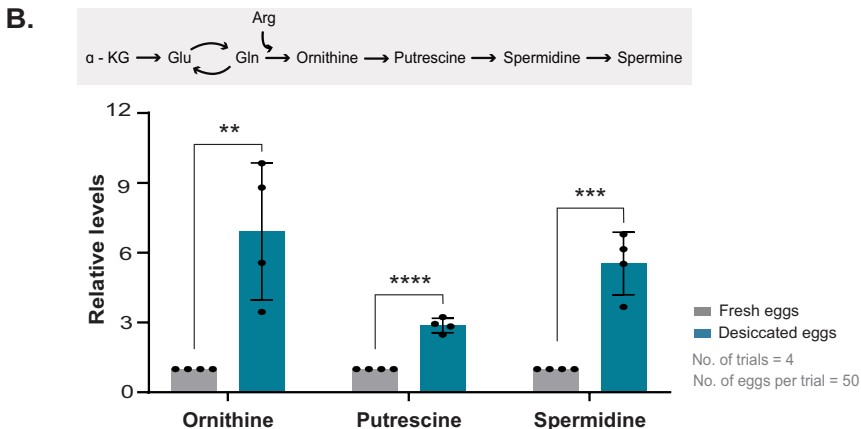

**Fig 3. Desiccated *Aedes* eggs acquire a hypometabolic state with increased polyamines accumulation.** (A) Top: Schematic showing the experimental set up for metabolite extraction and estimation of steady-state metabolites in respective pathways. Relative metabolite levels were calculated from peak areas obtained for that metabolite [60]. Also see S1F Fig for an illustration of egg collection for metabolite extraction. Bottom: The graph shows relative steady-state levels of specific amino acids, glycolysis, and TCA cycle metabolites. G6P –glucose-6-phosphate, F6P –fructose-6-phosphate, 3PG– 3-phosphoglycerate, PEP–phosphoenolpyruvate, R5P –ribose-5-phosphate, α-KG–alpha ketoglutarate. Data is represented as mean ± SD. The number of trials = 3. The number of eggs used per trial = 50. Also see S3A–S3C Fig for other measured sugar phosphates, trehalose amounts, and amino acids. (B) Polyamine accumulation in desiccated eggs. (Top) Pathway of polyamine synthesis, derived from the upper arm of the TCA cycle and arginine metabolism. (Bottom) The graph represents relative amounts of ornithine, putrescine, and spermidine. Data is represented as mean ± SD. The number of trials = 3. The number of eggs used per trial = 50. Also see S3D Fig for polyamine levels in *An. stephensi* eggs. Statistical significance was calculated using an unpaired Student *t* test. *$p < 0.05$, **$p < 0.01$, ***$p < 0.001$, ns–no significant difference. Dataset underlying Fig 3A and 3B can be found in S4 Table.

which are derived from arginine metabolism, substantially accumulate during desiccation. Contrastingly, some other desiccation–tolerant organisms such as yeast, the dauer stage of *C. elegans* or the larvae of *P. vanderplanki* accumulate trehalose [11–14,16,17], while tardigrades accumulate IDPs in response to desiccation [20]. Interestingly, the desiccation–tolerant dauer larvae of *C. elegans* also up-regulate polyamine biosynthesis [12]. All of these molecules can function to protect nucleotides, proteins, and membranes from damage due to water loss. Hence, there appears to be a logic to the metabolic rewiring leading to the production of these molecules.

## Polyamines are essential for the desiccation tolerance of *Ae. aegypti* eggs

Studies from the dauer stage of *C. elegans* larvae found that the polyamine-producing enzymes ornithine decarboxylase (OCD) and spermidine synthase increased during desiccation [12], suggesting that polyamines might enable desiccation tolerance. In general, polyamines have pleiotropic "protective" roles, by binding to nucleic acids, proteins, and membrane phospholipids, can also act as ROS scavengers and chemical chaperones, and form liquid crystals, all of which are useful for desiccation tolerance [61,62].

Since the desiccated *Aedes* eggs specifically accumulated polyamines, we hypothesised that polyamines might assist in desiccation tolerance. Since polyamines have essential roles in cells [61], we adopted an inhibitor-based experimental setup to test their importance in desiccation tolerance (S4A Fig). We used difluoromethylornithine (DFMO) to specifically inhibit OCD, a rate-limiting enzyme in polyamine biosynthesis [63,64]. The drug dosage was titrated and optimised such that a sublethal concentration of 1 mM was used in the blood feed, to ensure that the survival of the adult mosquito or its ability to lay eggs was not significantly altered. We obtained eggs from mosquitoes that were blood-fed with and without DFMO and subjected these eggs to desiccation, followed by rehydration and assessed viability (S4A Fig). We observed significantly reduced hatching in desiccated eggs obtained from females treated with the inhibitor, while fresh eggs that were treated with the inhibitor (but not subject to desiccation) retained a high percentage of hatching similar to untreated controls (Fig 4A). Consistently, we measured the levels of polyamines in fresh as well as desiccated eggs obtained from DFMO-treated and untreated mosquitoes. We observed reduced polyamines in desiccated eggs that were treated with DFMO in contrast to desiccated eggs from the untreated batch (S4B Fig). Collectively, these data suggest that the accumulation of polyamines is necessary for desiccation tolerance of *Aedes* eggs and reducing polyamine biosynthesis renders eggs sensitive to desiccation.

## Polyamine synthesis and lipid breakdown function synergistically to enable *Aedes* egg desiccation tolerance and larval hatching upon rehydration

Next, we assessed the importance (for desiccation and revival) of the observed increase in fatty acid breakdown pathways, also correlating with altered lipid reserves in desiccated *Aedes* eggs. The need for fatty acid breakdown for desiccation tolerance was not immediately apparent. We considered 2 scenarios: one where fatty acid breakdown was critical for desiccation tolerance and sustenance of the pharate larvae during the desiccation phase, and the second, more nuanced possibility where fatty acid breakdown was required to fuel energy metabolism post rehydration, thereby enabling survival post reentry of water. To investigate the role of fatty acid oxidation, we inhibited lipid oxidation using 2-bromopalmitic acid (2-BPA), a well-studied carnitine acetyltransferase inhibitor [65,66] and ensured that a sublethal dose of 0.8 mM was used, with controls similar to those described earlier for polyamine inhibition (S4A and S4B Fig). Concentrations of 2-BPA above 1.2 mM resulted in decreased adult mosquito

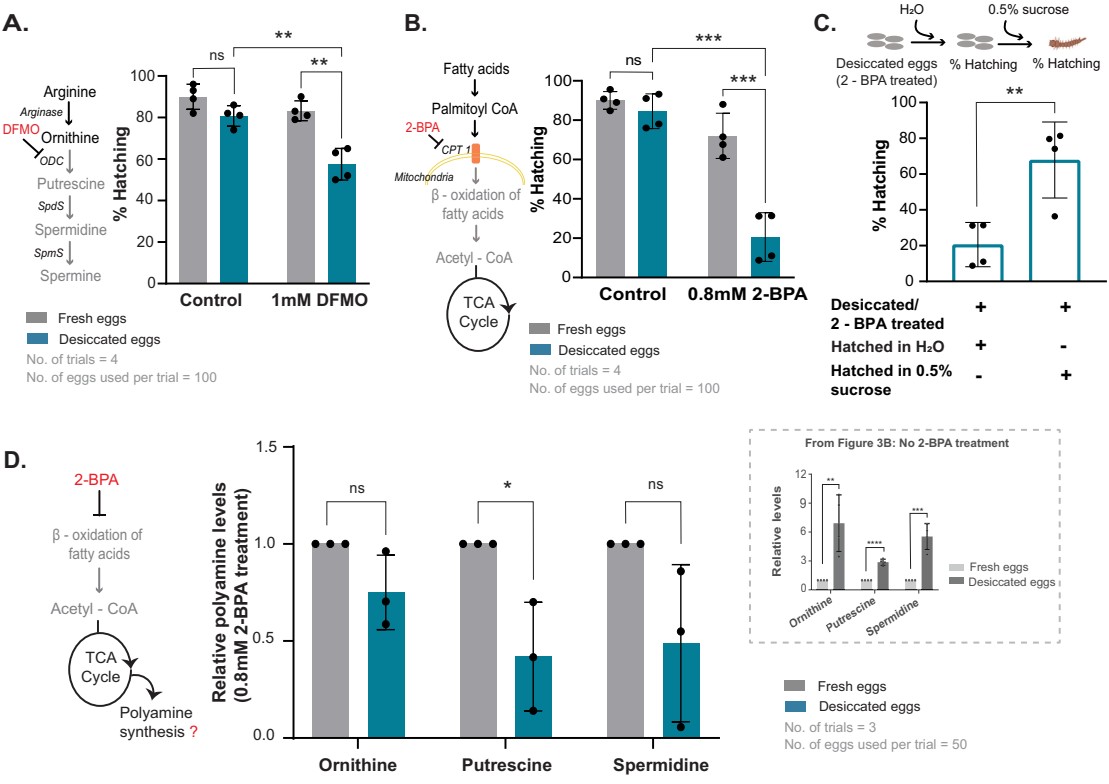

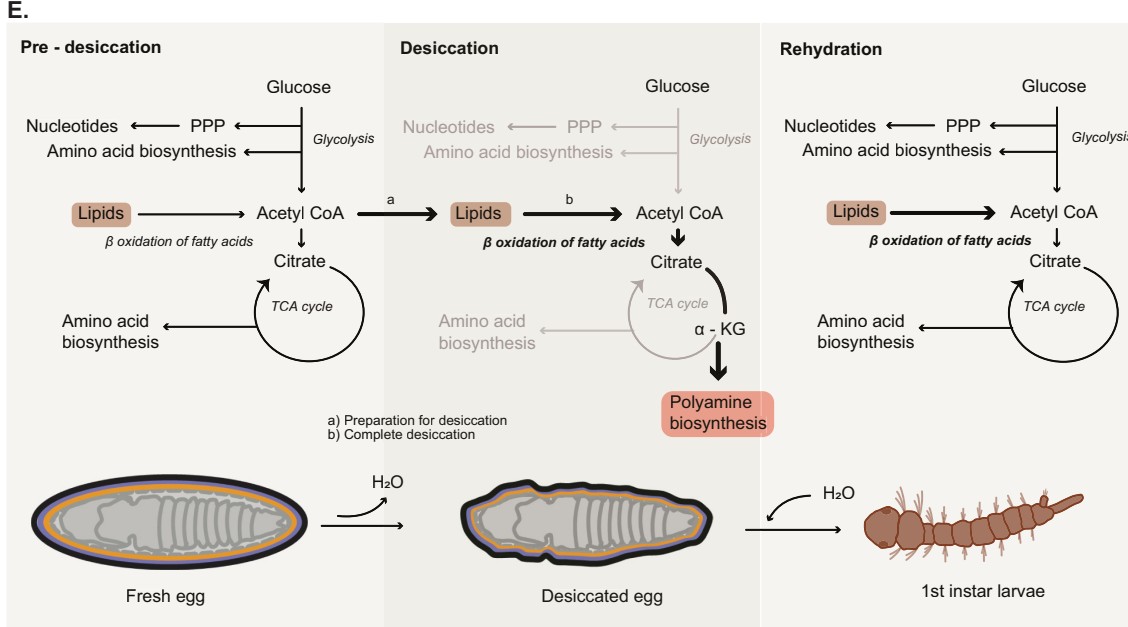

**Fig 4. Polyamine synthesis and lipid breakdown function synergistically to enable *Aedes* egg desiccation tolerance and larval hatching post rehydration.** (A) Desiccation tolerance in *Aedes* eggs and dependence on polyamines. The schematic on the left shows the inhibition of ODC by DFMO. Inhibiting ODC should lead to reduced putrescine, spermidine, and spermine (shown in grey) and the accumulation of ornithine (shown in black). The graph shows the reduction in the percentage of hatching of eggs post desiccation when blood-fed mosquitoes were treated with DFMO in contrast to desiccated eggs that were obtained from untreated mosquitoes. The percentage hatching was compared between the following groups: control fresh eggs versus control desiccated eggs, treated fresh eggs versus treated desiccated eggs, and control desiccated eggs versus treated desiccated eggs. DFMO—difluoromethylornithine, ODC—ornithine decarboxylase, SpdS—spermidine synthase, SpmS—spermine synthase. Data is represented as mean ± SD. The number of trials = 4. The number of eggs used per trial = 100. Also see S4A Fig for a schematic of the experimental set for inhibitor

assays and S4B Fig for polyamine levels after inhibitor treatment. (B) Inhibiting fatty acid oxidation and tolerance to desiccation in *Aedes* eggs. The schematic on the left depicts the mode of action of 2-bromopalmitic acid (2-BPA) that inhibits carnitine palmitoyl transferase 1 (CPT 1), a rate controlling enzyme for β-oxidation of fatty acids in the mitochondria. The graph shows the percentage of fresh and desiccated eggs hatching into first instar larvae when adult mosquitoes were fed with DMSO (control) and 2-BPA. The percentage hatching was compared between the following groups: control fresh eggs versus control desiccated eggs, treated fresh eggs versus treated desiccated eggs, and control desiccated eggs versus treated desiccated. Data is represented as mean ± SD. The number of trials = 4. The number of eggs used per trial ≥ 100. Also see S4C Fig for lipid levels after inhibitor treatment. (C) Requirement of stored lipids for recovery upon rehydration. The top schematic shows the experimental setup, where the hatching of desiccated mosquito eggs treated with 2-BPA and hatched in water, was compared to that of desiccated eggs treated with 2-BPA and hatched in 0.5% sucrose. The graph shows the percentage hatching of eggs to first instar. Data is represented as mean ± SD. The number of trials = 4. Also see S4D Fig for a consolidated schematic on the effects of 2-BPA inhibition on lipid metabolism during desiccation and rehydration in *Aedes* eggs. (D) Requirement of lipid breakdown for polyamine accumulation in desiccated eggs. The graph represents steady-state levels of polyamines—ornithine, putrescine, and spermidine in fresh and desiccated eggs under 2-BPA treated conditions. Data is represented as mean ± SD. The number of trials = 3. The number of eggs used per trial = 50. The inset within the dashed-line box reproduces Fig 3B, for comparison, representing changes in polyamines upon desiccation, without 2-BPA treatment. (E) Model illustrating the metabolic rewiring in response to desiccation in *Aedes* embryos. During desiccation, proteins and metabolites of the key energy-producing pathways such as glycolysis and TCA cycle (lower arm) reduce and there is an increase in polyamine biosynthesis and lipid breakdown. Eggs sense desiccation and use lipid metabolism as a strategy to prepare them for the dormant state, including diversion of resources towards polyamine synthesis. These fatty acid reserves are also utilised as an energy source for rapid reactivation of metabolism upon rehydration. The polyamines protect the egg during the dormant state. Thick black arrows indicate pathways up-regulated during desiccation and grey arrows indicate the pathways down-regulated during desiccation. Statistical significance was calculated using an unpaired Student *t* test. *$p < 0.05$, **$p < 0.01$, ***$p < 0.001$, ns–no significant difference. Raw data underlying Fig 4A–4D can be found in S4 and S5 Tables.

viability or egg laying and were avoided. 2-BPA treatment significantly reduced the viability of desiccated eggs, as observed by a sharp decline in hatching (Fig 4B). In contrast, fresh eggs from 2-BPA-treated mosquitoes hatched normally. Since desiccated eggs, as well as the first instar larvae hatching from desiccated eggs, had lower lipid levels, we next hypothesised that lipid catabolism might serve as a source of energy for rapid reactivation of metabolism upon rehydration. If this were so, when an alternate, excellent energy source is provided to desiccated eggs during rehydration, it should rescue survival. To test this possibility, we used the same experimental system as above, where eggs from mosquitoes treated with 2-BPA were desiccated. However, to these eggs, we provided an alternate, high-energy food source (0.5% sucrose) during rehydration, and then estimated egg hatching. When the inhibitor-treated desiccated eggs were rehydrated in the presence of sucrose, we observed near-complete rescue of hatching (Fig 4C). These data collectively suggest that a primary requirement of increased lipid breakdown in desiccated *Aedes* eggs would be to fuel energy production by providing required precursors for energy metabolism, post rehydration.

Finally, we asked if the fatty acid oxidation and lipid catabolism in desiccated eggs was itself coupled to increased polyamine biosynthesis, as part of a comprehensive desiccation-specific metabolic program. The reasoning for this comes from the knowledge that polyamine biosynthesis (which is derived from α-ketoglutarate and arginine biosynthesis) will require a steady supply of acetyl-CoA. When glycolysis decreases (as observed in desiccated *Aedes* eggs), lipid breakdown could serve as an alternate source of acetyl-CoA. In such a scenario, inhibiting lipid oxidation should reduce or prevent polyamine accumulation in desiccated eggs. To test this idea, we used 2-BPA-treated mosquito eggs (in the approach described earlier) and now quantified polyamines in fresh and desiccated eggs. Polyamine amounts reduced or remained unchanged in desiccated eggs treated with 2-BPA (Fig 4D). Note that, when not treated with the inhibitor, desiccated *Aedes* eggs increase polyamine levels (Fig 4D, inset). These data suggest that fatty acid oxidation and polyamine synthesis pathways are coupled in this desiccation program, with fatty acid oxidation being utilised to maintain the increase of polyamines.

In summary, we uncover a unique, protective metabolic program in *Ae. aegypti* eggs in response to complete desiccation (Fig 4E). In order to survive desiccation, the embryos must first reach an advanced developmental stage only after which their development is arrested as

pharate larvae inside the egg, where a sudden drop in humidity is sensed by the embryo to prevent larval hatching. While mosquito eggs have an obvious cuticle that confer limited protection to the developing larva, more importantly, the larva within the eggs undergo a precise proteomic and metabolic rewiring that enables resistance to desiccation as well as survival post rehydration. When exposed to drying conditions, the developing larva within the eggs alter their proteomes towards a metabolic state where "energy and growth" metabolism, along with associated oxidative steps (for example, with glycolysis and the later part of the TCA cycle) are reduced resulting in a hypometabolic state. Upon sensing desiccation, the acetyl-CoA accumulated during the hydrated state is likely utilised to synthesise lipids. Concurrently, the existing carbon and nitrogen reserves are channelled towards the production of polyamines, which provides protective functions. As the desiccation phase progresses, the metabolic program shifts towards lipid breakdown which serves 2 functions. First, lipid breakdown is required to accumulate polyamines (which protect from desiccation). Second, lipid breakdown serves an essential role in enabling embryos to restore energy homeostasis and refuel recovery once the eggs are rehydrated and the larvae complete their hatching.

We note interesting parallels between our study (focussed on mechanisms of insect desiccation tolerance and recovery) and the phenomenon of embryonic diapause—a programmed developmental arrest, characterised by tolerance to various environmental stresses and substantial energetic changes [26]. During unfavourable conditions, female mosquitoes lay diapausing eggs, where embryonic development is complete, yet hatching is suspended [27]. Interestingly, diapausing eggs have higher lipids due to increased expression of *fatty acid synthase* and *lipid storage droplet protein 2* [56,67–69]. During the later stages, the expression of *lipases* and *hydrolases* increase, suggesting the possibility that lipids are being utilised as a possible energy source [67]. Diapausing embryos also show increased *pepck* expression, consistent with increased gluconeogenesis [56,67,70]. At this stage, any causal or critical role of metabolic rewiring during diapause remains unresolved. Given this convergence, our study suggests conserved principles of metabolic rewiring in these two phenomena. This allows a speculative hypothesis that the diapause state functions as a transition point towards either recovery or extreme dormancy, determined by the extent and nature of metabolic rewiring.

Our study broadly illustrates the nature of reorganisation of the metabolic network that collectively protects the developing larva from damage due to water loss, as well as enables recovery and intact hatching when water reenters the eggs. In particular, we uncover mechanisms to restore metabolism upon rehydration that are effective in a "semi-closed" system such as a mosquito egg (where exchange of nutrients is almost impossible), which hatches in a fresh water environment. Given the importance of the *Ae. aegypti* as a primary vector for numerous viral diseases (yellow fever, dengue, chikungunya, and others) that affects nearly half the world's population, as well as the rapid geographical expansion of this mosquito vector, we anticipate that this work will foundationally enable orthologous studies to reduce *Aedes* egg survival and global spread. Additionally, some of the inhibitors described here that reduce desiccation resistance in *Ae. aegypti* eggs, as well as new ones affecting other steps in the egg desiccation tolerance pathway, may prove useful as vector-control agents.

## Materials and methods

### Mosquito sources and rearing

*Aedes aegypti* and *Anopheles stephensi* were maintained and reared at the iBSL2 facility in the insectary at DBT-inStem and the Tata Institute for Genetics and Society. Adults were fed on 8% sucrose, 2% glucose, a multivitamin solution (Polybion SF Complete, Merck), and 0.2% methyl paraben; 10-day old females were blood-fed with human O+ve blood obtained

commercially from a blood bank using the Hemotek membrane feeding system. Larvae were reared in rectangular trays and fed on a liquid diet consisting of a mixture of dog biscuits and yeast. Appropriate approvals from the institutional biosafety committee of DBT-inStem were obtained and our studies adhered to the guidelines set forth by the human ethics as well as bio-safety committees.

## Synchronous egg laying

Female mosquitoes lay eggs 3 to 4 days post a blood meal, and 10 gravid females were chosen randomly from cages and transferred to plugged tubes containing moist cotton and lined with moist filter paper. Tubes were placed in a humid, light-protected chamber at 28°C. In all cases, oviposition lasted for 1 h. The eggs were allowed to complete embryonation by placing the filter papers on moist cotton for 48 h (Fig 1A). These eggs (48 h old) were termed as "fresh eggs."

## Desiccation assay for the eggs of *Aedes* and *Anopheles*

Fresh eggs of *Ae. aegypti* and *An. stephensi* were collected as described above. These eggs were divided into 8 batches. One of the batches was not subjected to desiccation (termed as 0 days post desiccation in Fig 1A and 1C) and immediately placed in water. The other 7 batches were desiccated for 3, 6, 9, 12, 15, 18, and 21 days by placing them on Whatman filter paper. The desiccated eggs were then rehydrated by placing the filter paper (containing eggs) in RO water. The eggs hatched into first instar larvae within 1 h and the larvae are counted (S1A Fig). The percentage hatching was calculated as follows:

$$\% \text{ Hatching} = \frac{\text{Total number of hatched 1st instar larvae} \times 100}{\text{Total number of eggs}}$$

The morphological details of fresh and 21 days desiccated eggs and the first instar larvae hatching from them were observed under a Nikon SMZ18 stereomicroscope.

## Mosquito egg clarification for morphological study

Synchronised mosquito eggs were obtained as described above. Fresh and 21 days desiccated eggs were fixed and clarified as described by Trpiš [71] to make the egg shells transparent enabling morphological analysis. Briefly, eggs were transferred to 1.5 ml tubes fixed in FAA solution (10% formalin, 5% glacial acetic acid, and 50% ethanol in water) for 30 min. After 30 min, the sample was washed thrice in 1× phosphate-buffered saline (PBS). This was followed by addition of Trpiš clarification solution to the eggs and incubation at 4°C for 12 h. The bleached eggs were then washed again with PBS and transferred onto glass slides. The eggs were visualised under a 1× objective in a Nikon SMZ18 stereomicroscope.

## Acquisition of embryonic resistance to desiccation (ERD)

Synchronised eggs of *Aedes* were collected as described above. Eggs were kept at 27°C on moist cotton until the required age, the onset being considered the end of the 1-h egg laying period. Embryonic age was assigned as hours after egg laying (HAE). At distinct embryogenesis time points (0 HAE, 4 HAE, 8 HAE, 15 HAE, 24 HAE, and 48 HAE) replicates consisting of 100 eggs each were transferred to a dry Whatman No.1 filter paper and desiccated for 10 days. Egg viability was assessed by placing each of the filter papers containing eggs in RO water and counting the first instar larvae hatching from it.

### *Aedes* egg collection for various assays

Fresh eggs were collected as described above and divided into 2 batches. One batch always stayed hydrated and was not subject to desiccation. The other batch, termed as "desiccated eggs" was dried for a total period of 21 days (S1F Fig). Required amount of fresh and desiccated eggs were used for all the developmental assays and egg extract preparation for proteomic, metabolomics, and other biochemical assays as described in detail below.

### Development of *Ae. aegypti* emerging from fresh and desiccated eggs

Larval development was quantified in artificial containers. Synchronous hatching of fresh and desiccated eggs was induced for 1 h. First instar larvae were placed in trays containing 1.5 litres of RO water, and 6 ml of 2% larval food slurry was added for the first instars. Second instars were fed on 10 ml of 2% slurry. For the third and fourth instar larvae, 0.5 g of larval food was added to 2 litres of RO water. Each tray contained up to 150 larvae. Larval development was quantified in terms of length, as measured daily from the day of hatching till the day of pupation. Additionally, performance across conditions was also evaluated by measuring the time taken to pupate (days since hatching) and the time to eclose into adults (days since pupation). Percentage hatching, pupation, and eclosure were estimated for mosquitoes emerging from fresh or desiccated eggs.

### Proteomics

**Sample preparation.** A total of 150 fresh and desiccated eggs of *Aedes aegypti* were collected as described above. All the 150 fresh eggs were pooled and transferred to clean 1.5 ml Eppendorf tubes and crushed on ice using a micro-pestle in 100 μl of 1× PBS with 3× protease inhibitor cocktail (Sigma Aldrich, P8340). A similar procedure was followed for the collected desiccated eggs. For protein extraction from first instar larvae, synchronous hatching from fresh and desiccated eggs was induced for 1 h. Approximately 300 hatched larvae were crushed in 1× PBS containing 3× protease inhibitor cocktail (Sigma Aldrich, P8340) supplemented with 10 mM PMSF, 10 mM iodoacetamide, and 4 mM EDTA. The samples were sonicated for 5 min (5 s pulse and 3 s rest). The supernatant was collected after centrifuging at 13,000 RPM (10 mins, 4°C) and total protein was estimated by BCA (Pierce BCA Protein Assay Kit, Thermo Fisher Scientific, 23225). The supernatant was mixed with 1× Laemmli buffer and boiled at 95°C for 10 min, and 10 μg of each sample was resolved on a 4% to 12% precast gradient gel (Invitrogen, NP0322BOX) by standard electrophoresis. The gel was stained using Coomassie Brilliant Blue G– 250 (Thermo Fisher Scientific, LC6060) for 1 h, destained, and imaged using the iBright imaging system (Thermo Fisher Scientific). Each lane was cut into 4 slices and was separately in gel digested with trypsin, extracted and vacuum dried according to the standard protocol described by Shevchenko and colleagues [72].

**LC-MS/MS.** Samples were analysed on Thermo Orbitrap Fusion Tribrid mass spectrometer coupled to a Thermo Nano-flow liquid chromatography system (EASY-nLC 1200 series). Dried digests were reconstituted in 2% acetonitrile/0.1% formic acid and 0.3 μg injected onto a LC pre-column (Thermo Fisher Scientific Acclaim Pep map 100, 75 μm × 2 cm, Nanoviper C18, 3 μm, 100 Å) for separation followed by loading onto the column (Thermo Fisher Scientific Easy Spray Pep map, RSLC C18 3 μm, 50 cm × 75 μm, 100 Å) at a flow rate of 300 nL/min. Solvents used were: 0.1% formic acid (Buffer A) and 80% acetonitrile + 0.1% formic acid (Buffer B). Peptides were eluted using a gradient from 10% to 95% of Buffer B for 60 min. Full scan MS spectra (from m/z 375 to 1,700) were acquired at a resolution of 120,000. Precursor ions were sent for subsequent fragmentation by HCD at a collision energy of 32%. MS and

MS/MS data were obtained in the orbitrap (Thermo Orbitrap Fusion Tribrid MS, Thermo Fisher Scientific).

**Data analysis.** Data analysis was performed using proteome discoverer (version 2.1). The resulting MS/MS data was searched against the *Aedes aegypti* database (Taxonomy id: 7159, 36,032 sequences and 19,407,208 residues). Trypsin was the enzyme used and two missed cleavages were allowed. Searches were performed using a peptide mass tolerance of 10 ppm and a product ion tolerance of 0.6 Da resulting in a 1% false discovery rate. A comparison of the identified peptides in desiccated and fresh eggs was done and the data was organised into categories of (i) increased post desiccation (ii) found equally in fresh and desiccated eggs and (iii) decreased post desiccation based on their emPAI scores. Uncharacterised proteins were assessed using PFAM (http://pfam.xfam.org/) for known protein domains and functions were manually assigned (see S1 Table for a complete list of proteins increasing or decreasing post desiccation in *Aedes* eggs). Gene ontology analysis was carried out using VectorBase (https://vectorbase.org/vectorbase/app/). GO terms with corrected *p*-value <0.05 (Benjamini correction) were considered significantly enriched (see S2 Table for list of all the enriched GO terms). The mass spectrometry-based proteomics data have been deposited to the ProteomeXchange Consortium via the PRIDE [73] partner repository, with the data set identifier PXD044525.

## Metabolomics

**Sample preparation.** A total of 50 fresh and desiccated eggs were collected as described previously, transferred to Eppendorf tubes, and washed in 80% ethanol (extraction buffer). Eggs were then crushed in 450 μl of extraction buffer. Samples were heated for 10 min at 85˚C and immediately placed on ice. The samples were spun at 13,000 rpm for 10 min at 4˚C. The supernatant was collected into fresh tubes and divided into 2 parts of 200 μl and 1 part of 20 μl. The parts containing 200 μl of the extract were dried using a speed vac.

**OBHA derivatization.** Derivatization in order to detect carboxylic acids was done modifying methods described before [60] using the 20 μl part. The derivatized extract was dried using a speed vac.

**LC-MS/MS and data analysis.** Steady-state levels of metabolites were analysed using methods described earlier [60]. Briefly, extracted metabolites were separated using a Synergi 4-μm Fusion-RP 80 Å (150*4.6 mm, Phenomenex) LC column on Shimadzu Nexera UHPLC system, using 0.1% formic acid in water (Solvent A) and 0.1% formic acid in methanol (Solvent B) for amino acids, nucleotides and TCA metabolites and 5 mM ammonium acetate in water (Solvent A) and 100% acetonitrile (Solvent B) for sugar phosphates. The flow parameters were as described in [60]. Data was acquired using an AB Sciex Qtrap 5500 and analyzed using the Analyst 1.6.2 software (Sciex). Amino acids and TCA intermediates were detected in positive polarity while sugar phosphates were detected in negative polarity mode. The parent and daughter ion *m/z* parameters for metabolites are given in the S3 Table. The area under the curve for obtained peaks was obtained using Multi Quant (Version 3.0.1). Analysed data were normalised and plotted (see S4 Table for peak areas of all the detected compounds).

## Trehalose measurements from yeast and mosquito egg samples

A total of 10 mg of fresh and desiccated *Aedes*' eggs were transferred to Eppendorf vials; 250 μl of 0.25 M sodium carbonate was added to all the samples and crushed using a micro-pestle, and 10 mg of yeast cell pellet (harvested during logarithmic phase of growth) was used for the trehalose assay. Approximately 250 μl of 0.25 M sodium carbonate was added to the cell pellet and all the samples were boiled at 95˚C. Enzymatic measurement of trehalose in yeast cells and

mosquito eggs was performed according to the protocol described in [59,74]. Absorbance at 540 nm was determined and compared with the glucose standard to assess the quantity of glucose liberated from trehalose.

### Estimation of total lipids in mosquito eggs and larvae

Total lipids in mosquito eggs and larvae were determined by extraction with 1:1 chloroform methanol followed by a reaction with $H_2SO_4$ and phospho-vanillin reagent as described by [75]. Briefly, samples were crushed in 70 μl chloroform: methanol (1:1) and the supernatant was used. Lipid standards of 0 to 500 μg were prepared such that the final volume was 50 μl. The samples were heated at 60°C so that the solvents evaporated completely, and 20 μl of concentrated $H_2SO_4$ was added to the standards as well as the samples and heated at 100°C for 10 min. Samples were brought to room temperature and 480 μl of phospho-vanillin reagent was added to the tubes and incubated at 37°C in the dark for 10 min for colour development, and absorbance was measured at 530 nm.

### Inhibition of polyamine synthesis in mosquito eggs

In accordance with a previous study [76], we chose a concentration range (0 to 2 mM) for DFMO (Sigma Aldrich, D193) and a dose titration was carried out in blood and fed to 10-day-old adult mosquitoes ensuring that a sub lethal concentration of the drug was used for the assay. We subsequently administered 1 mM DFMO to adult mosquitoes. Note that higher doses of DFMO (>2 mM) led to reduced fertility and fecundity in female mosquitoes. Control groups were fed with the same volume of distilled water in blood. Fresh and desiccated eggs from control and inhibitor-treated mosquitoes were obtained as described earlier and the percentage hatching in these groups was calculated. Polyamine levels were estimated by mass spectrometry according to the method described above.

### Inhibition of β-oxidation of lipids

A similar titrated inhibitor-based approach as described above was followed to inhibit fatty acid oxidation in fresh and desiccated *Aedes* eggs, and 0.8 mM of 2-bromopalmitic acid (Sigma Aldrich, 238422) was added to fresh blood used for feeding 10 days old female mosquitoes. This concentration was specifically chosen based on our optimised range for DFMO treatment which served as a reference and from published data by [77]. Higher doses of the drug (>1.2 mM) led to the mortality of adult females. Control groups were fed with same volume of DMSO in blood. Fresh and desiccated eggs were collected as described earlier from control and inhibitor-treated mosquitoes. Percentage of eggs hatching was calculated. The percentage hatching of inhibitor-treated desiccated eggs was further checked 24 h post-revival with 0.5% sucrose. Total lipids were estimated in fresh and desiccated eggs obtained from the control and treated mosquitoes using the sulfo-phospho-vanillin method described above.

### Data visualisation and statistics

All the bar graphs were plotted using GraphPad Prism 8.4.2. Unpaired Student's *t* test was used to calculate statistical significance. *P*-values have been specified in the corresponding figure legends. Heatmaps were generated by the software Heatmapper (http://www.heatmapper.ca/). Selected GO terms were visualised in a bubble plot generated using enrichplot package of R. All the images/clip-art within the figure panels are original and were drawn by hand.

## Supporting information

**S1 Fig. Desiccation and larval development.** (A) Detailed schematic depicting *Aedes* and *Anopheles* egg desiccation assay. Synchronised eggs (1 h old) were collected 5 days post blood meal in plugged tubes containing moist cotton. These eggs were transferred onto fresh moist cotton to allow embryonation for 48 h. One batch of eggs stayed hydrated and hatched after 48 h (fresh eggs/0 days post desiccation). After 48 h, other batches were subsequently desiccated for 3, 6, 9, 12, 15, 18, and 21 days. Desiccated eggs were rehydrated by transferring them to trays containing water. First instar larvae hatching from fresh or desiccated eggs were counted to calculate the percentage hatching. Note: Desiccation assay was also performed in *An. stephensi* eggs using the same procedure described above. (B) Embryo structure and morphology. Clarified fresh (left–top and bottom) and desiccated eggs (right–top and bottom) were viewed under a stereo zoom phase contrast microscope to observe embryo morphology. (C) The table shows the duration taken by first instar larvae hatching from fresh or desiccated eggs to develop into pupae, and the duration that pupae take to eclose into adults. (D) Desiccation and larval development. The graph shows the percentage of fresh and desiccated eggs hatching into first instars, the percentage of larvae developing into pupae and the percentage of pupae developing into adult mosquitoes. Data is represented as mean ± SD. The number of trials = 5. The number of eggs/larvae used per trial = 150. The underlying raw data for this figure can be found in S5 Table. (E) Whole protein extract from first instar larvae hatching from fresh and desiccated eggs analysed on a Coomassie stained SDS-PAGE gel. Note: no overt differences in the band pattern in larvae emerging from fresh and desiccated eggs can be observed. The number of trials = 2 (1 and 2–2 trials of first instar larvae from fresh eggs, 3 and 4–2 trials of first instar larvae from desiccated eggs). The number of larvae used per trial approximately 300. Note: For each lane in the gel, approximately 300 larvae were pooled and lysed together for protein extraction. (F) Schematic depicting *Aedes* egg collection for various assays. Synchronised eggs were collected and divided into 2 batches. One batch (fresh eggs) was kept hydrated and not subject to desiccation. The other batch, termed as "desiccated eggs" was dried for a total period of 21 days. Extracts from both fresh and desiccated eggs were prepared for various experiments as detailed in the Materials and methods. Statistical significance was calculated using an unpaired Student $t$ test. $^*p < 0.05$, $^{**}p < 0.01$, $^{***}p < 0.001$, ns—no significant difference.
(EPS)

**S2 Fig. GO-based grouping of proteins that change during desiccation.** (A) Gene ontology (GO)-based analysis and grouping of proteins into functional categories. The bubble plot shows GO analysis of proteins up-regulated in desiccated eggs (black), equally expressed in fresh and desiccated eggs (grey) and proteins down-regulated in desiccated eggs (light grey). The rich factor indicated in the y-axis was calculated as the ratio of number of proteins annotated in a particular GO term to the total number proteins in that GO term. The colour of each bubble represents the corrected $p$-values (Benjamini correction) of each term involved in the analysis. The size of each bubble represents the number of proteins identified in this study belonging to the specific GO term. S2 Table lists all the enriched GO terms.
(EPS)

**S3 Fig. Additional metabolite measurements in fresh and desiccated eggs.** (A) Steady-state levels of additional glycolytic and PPP intermediates in *Aedes* eggs. The graph represents relative steady-state levels of G3P –glyceraldehyde-3-phosphate, F16BP–fructose-1,6 bisphosphate, S7P –sedoheptulose-7-phosphate. Data is represented as mean ± SD. The number of trials = 3. The number of eggs used per trial = 50. (B) Trehalose amounts in *Aedes* eggs before and after

desiccation. The graph represents relative trehalose levels between fresh and desiccated eggs and equal biomass of yeast. The number of trials = 3. Quantity of eggs or yeast used per trial = 10 mg. (C) Steady-state levels of all amino acids in fresh and desiccated *Aedes* eggs. The graph represents relative levels of amino acids. Data is represented as mean ± SD. The number of trials = 3. The number of eggs used per trial = 50. (D) Polyamine levels in the eggs of *An. stephensi*, a desiccation sensitive species. The graph represents relative steady-state levels of ornithine, putrescine, and spermidine. Data is represented as mean ± SD. The number of trials = 2. The number of eggs used per trial = 50. Statistical significance was calculated using an unpaired Student *t* test. $^*p < 0.05$, $^{**}p < 0.01$, $^{***}p < 0.001$, ns–no significant difference. Datasets for S3A–S3D Fig is provided in S4 and S5 Tables.
(EPS)

**S4 Fig. Additional metabolite measurements in inhibitor-treated eggs that undergo desiccation.** (A) An illustration showing the experimental setup for inhibiting ODC using DFMO or fatty acid oxidation using 2-BPA. Mosquitoes were fed with blood containing the inhibitor or the vehicle ($H_2O$ or DMSO, respectively). Desiccation assay was performed as described earlier with the fresh and desiccated eggs obtained from the control and inhibitor-fed mosquitoes. (B) Polyamine amounts in *Aedes* eggs under control and inhibitor (DFMO)-treated conditions. The graphs (i–iii) represent steady-state levels of polyamines—ornithine, putrescine, and spermidine in fresh and desiccated eggs under $H_2O$ (control) and DFMO-treated conditions. Polyamine levels were compared between the following groups: control fresh eggs versus control desiccated eggs, treated fresh eggs versus treated desiccated eggs, and control desiccated eggs versus treated desiccated eggs. Data is represented as mean ± SD. The number of trials = 4. The number of eggs used per trial = 50. (C) 2-BPA treatment for inhibiting beta-oxidation of fatty acids and lipid levels. The graph represents relative lipid levels in fresh and desiccated eggs under DMSO (control) and 2-BPA-treated conditions. Lipid levels were compared between the following groups: control fresh eggs versus control desiccated eggs, treated fresh eggs versus treated desiccated eggs, and control desiccated eggs versus treated desiccated eggs. Data is represented as mean ± SD. The number of trials = 4. The number of eggs used per trial = 50. (D) Schematic showing the consequences of inhibiting fatty acid oxidation in *Aedes* eggs. During desiccation, the stored fats are broken down and feed into the TCA cycle providing energy for the pharate larvae to hatch post rehydration (a). The percentage of eggs surviving desiccation reduces after fatty acid oxidation inhibition (b). When the eggs are rehydrated in 0.5% sucrose, sucrose serves as an alternate source of energy to sustain the hatching of desiccated eggs (c). Statistical significance was calculated using an unpaired Student *t* test. $^*p < 0.05$, $^{**}p < 0.01$, $^{***}p < 0.001$, ns–no significant difference. Data underlying S4, S4B, and S4C Fig can be found in S4 and S5 Tables.
(EPS)

**S1 Table. List of proteins identified by mass spectrometry.**
(XLSX)

**S2 Table. List of enriched Gene Ontology (GO) terms.**
(XLSX)

**S3 Table. Parent and daughter ion m/z parameters for reported metabolites.**
(XLSX)

**S4 Table. Peak area data of all the reported metabolites.**
(XLSX)

**S5 Table. Raw data numerical values underlying Figs 1–4 and S1–S4.**
(XLSX)

## Acknowledgments

We thank all the present and past members of SL and BB lab for useful discussions. We thank the NCBS/inStem campus mass spectrometry facility for instrument access and support. We thank Khushboo Agrawal for early help with mosquito rearing. We acknowledge the support from Dr. Sunita Swain (TIGS) and the entire insectary team at DBT-inStem and TIGS-CI. We thank Dr. Suresh Subramani for valuable comments on the manuscript.

## Author Contributions

**Conceptualization:** Baskar Bakthavachalu, Sunil Laxman.

**Data curation:** Anjana Prasad, Sreesa Sreedharan.

**Formal analysis:** Anjana Prasad, Sreesa Sreedharan.

**Funding acquisition:** Baskar Bakthavachalu, Sunil Laxman.

**Investigation:** Anjana Prasad, Sreesa Sreedharan, Baskar Bakthavachalu, Sunil Laxman.

**Methodology:** Anjana Prasad, Sreesa Sreedharan, Baskar Bakthavachalu, Sunil Laxman.

**Project administration:** Baskar Bakthavachalu, Sunil Laxman.

**Resources:** Baskar Bakthavachalu, Sunil Laxman.

**Supervision:** Baskar Bakthavachalu, Sunil Laxman.

**Validation:** Anjana Prasad, Sreesa Sreedharan.

**Visualization:** Anjana Prasad, Sreesa Sreedharan, Baskar Bakthavachalu, Sunil Laxman.

**Writing – original draft:** Anjana Prasad, Baskar Bakthavachalu, Sunil Laxman.

**Writing – review & editing:** Anjana Prasad, Sreesa Sreedharan, Baskar Bakthavachalu, Sunil Laxman.

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
