## [Editor Report · Decision Letter 0]

26 Apr 2023

Dear Dr Laxman, 

Thank you for submitting your revised Review Commons manuscript entitled "Aedes aegypti eggs use rewired polyamine and lipid metabolism to survive desiccation" for consideration as a Short Report by PLOS Biology.

Your manuscript has now been evaluated by the PLOS Biology editorial staff, as well as by an academic editor with relevant expertise, and I'm writing to let you know that we would like to send your submission out for re-review.

IMPORTANT: The Academic Editor noted that a key aspect of the expertise was missing from the two Review Commons reviewers, namely direct familiarity with mosquito biology, so we will be inviting one further reviewer to assess your revised manuscript; as a result, further significant reviewer requests may emerge. Also, while you've submitted this as a Short Report, we think that a Discovery Report may be the more appropriate article type; no re-formatting is needed, but please change your article type to "Discovery Report" when you upload your additional metadata (see next paragraph).

Once your full submission is complete, your paper will undergo a series of checks in preparation for re-review. After your manuscript has passed the checks it will be sent out for review. To provide the metadata for your submission, please Login to Editorial Manager (https://www.editorialmanager.com/pbiology) within two working days, i.e. by Apr 28 2023 11:59PM.

Kind regards,

Roli Roberts

Roland Roberts, PhD

Senior Editor

PLOS Biology

rroberts@plos.org

---

## [Decision Letter · Decision Letter 1]

22 Jun 2023

Dear Dr Laxman,

Thank you for your patience while your revised Review Commons manuscript "Aedes aegypti eggs use rewired polyamine and lipid metabolism to survive desiccation" was peer-reviewed at PLOS Biology. It has now been evaluated by the PLOS Biology editors, an Academic Editor with relevant expertise, and by three independent reviewers. Reviewers #1 and #2 are the corresponding Review Commons reviewers. Unfortunately, reviewer #2 did not agree to re-review. In addition, because neither of these two reviewers was an expert on mosquitos, the Academic Editor requested that we seek further advice from reviewers with this expertise (reviewers #3 and #4).

You're see that reviewer #1 is now satisfied (and disagrees with reviewer #2's previous semantic point about anhydrobiosis). However, while both reviewers #3 and #4 find your study interesting, they each raise a number of concerns that must be addressed with ew experimental data (where applicable) before further consideration.

Specifically, reviewer #3 raises several methodological concerns (your decision to use larval length, and the specificity of your inhibitor assays); s/he also asks you to cite two papers and asks about ethics. Reviewer #4 was concerned that the experiment confounds aging and dehydration, and that a better control is required.

VERY IMPORTANT: The Academic Editor provided the following guidance, which you should read and comply with:

"What reviewer #3 recommends is important - and doable. I agree completely with his/her comments. It sounds like the authors have the data for comment 1. Comment 2 requires thought and deeper writing, certainly doable and also important. Comment 4 is a request to add info that they surely have. Comment 3 I think can be written around; inhibitors do have problems but they are also good for probing processes, at the stage that this work is in. Reviewer #4's first comment is absolutely valid but the authors' response to first-review that it is impossible to keep eggs in humid situations that long without their hatching is valid. However, they certainly could do the additional control that reviewer #4 suggests. His/her other comment like reviewer #3's: more depth of background, writing, and context; they should certainly do this."

In light of the reviews, which you will find at the end of this email, we would like to invite you to revise the work to thoroughly address the reviewers' reports.

Given the extent of revision needed, we cannot make a decision about publication until we have seen the revised manuscript and your response to the reviewers' comments. Your revised manuscript is likely to be sent for further evaluation by all or a subset of the reviewers.

**IMPORTANT - SUBMITTING YOUR REVISION**

*Re-submission Checklist*

*Published Peer Review*

*PLOS Data Policy*

*Blot and Gel Data Policy*

Sincerely,

Roli Roberts

Roland Roberts, PhD

Senior Editor

PLOS Biology

rroberts@plos.org

REVIEWERS' COMMENTS:

Reviewer #1:

The authors satisfactorily addressed most of my comments. In my view, the authors also satisfactorily addressed most of R2's concerns. I do not share R2's central concern on whether this study is really about surviving desiccation - viability is maintained in dried eggs hat -or something else. Respectfully, R2 is debating semantics and definitions that multiple people can debate forever without agreeing because one can always include or exclude a requirement for each definition that others would disagree with (e.g. desiccation avoidance versus desiccation tolerance). In the study, the eggs are left without any water, on Watman paper, in dry environments (i.e., relative vapor pressure is low). There may be some water left in the eggs. But at the macroscopic scale, it's fair to say that the eggs are dehydrated as the authors say. Hence, I disagree with R2's main criticism. 

I recommend a publication of this work in PLoS Biology.

Reviewer #2:

[did not re-review]

Reviewer #3:

This study focuses on understanding a key component of Aedes mosquito biology: the ability of eggs to hatch after desiccation. As the authors highlight, this desiccation resistance is a key to ability of these species to be transported and invade new locations and survive dry periods and is an important part what makes them so difficult to control. Overall, I think this is a very interesting study and worthy of publication. I have some concerns about an some aspects of the methodology. My comments will focus on aspects specifically related to mosquito biology.

1. It is unclear why larval length measured and how this would relate to development. The direct measure of larval development (how quickly L1 larvae develop into L2, L3, L4, Pupae and adults) demonstrated the same relationship, but the rationale for measuring length was not clear. The relationship between size of larvae and development can actually be inverse with smaller individual emerging from larvae that have developed more quickly. Unless there is a reason to present this data, I would remove it as it is not clear what it tells us about development. 

2. There is other work demonstrating the role of lipid metabolism in desiccation resistance during diapause. For example: Reynolds et al. 2012 Transcript profiling reveals mechanisms for lipid conservation during diapause in the mosquito, Aedes albopictus https://doi.org/10.1016/j.jinsphys.2012.04.013. Urbanski et al. 2010 The molecular physiology of increased egg desiccation resistance during diapause in the invasive mosquito, Aedes albopictus

Proc. R. Soc. B Biol. Sci., 277 (2010), pp. 2683-2692,-- These seem worth discussing in light of the protein data reported here. 

3. The inhibitor assays seem potentially problematic as interfering with lipid metabolism could have profound effects on female mosquito physiology including bloodmeal digestion and egg production. The study convincing demonstrates the differences between treatments, but it seems these could be a subset of many differences in these eggs. Can the authors provide additional explanation or rationale for how this manipulation is targeting only the aspects of egg metabolism that they intend to manipulate?

4. There needs to be an inclusion of the blood source (human? animal?) and appropriate ethical approvals for using blood included in the paper. 

Reviewer #4:

The goal of this study is to link polyamine rewiring and lipid metabolism with dehydration resistance in Aedes. The study is interesting and the processes of lipid metabolism and polyamine changes are likely involved in dehydration resistance (a more careful review of RNA-seq studies on dehydration in insects will show these aspects have been already hinted). 

Concern

1. The major concern that I have is that there is not a proper control to disentangle aging and dehydration. The control is fresh eggs (two days old) and the dehydrated in 21 days of age. So time and dehydration cannot be unlinked, making the conclusions linking dehydration and phenotypes observed as flawed in the current study. There are methods to allow the eggs to remain much more hydrated for extended periods (such as storage at high humidities), which would be a more appropriate control along side the 48 hour sample. These processes that have been identified are likely involved in dehydration, but the results are far from solid without the extra control. Importantly, we we store Aedes aegypti in the lab, we will start seeing a reduction in viability are 40-60 days at conditions described fro some lines, suggesting a very young to mid-life comparison. 

2. There is substantial literature mosquito (and other insects) dehydration tolerance from eggs to adults, which is not been discussed or reviewed that needs to be included.

---

## [Editor Report · Decision Letter 2]

7 Sep 2023

Dear Dr Laxman,

Thank you for your patience while we considered your revised manuscript "Aedes aegypti eggs use rewired polyamine and lipid metabolism to survive desiccation" for publication as a Discovery Report at PLOS Biology. This revised version of your manuscript has been evaluated by the PLOS Biology editors and the Academic Editor.

Based on our Academic Editor's assessment of your revision, we are likely to accept this manuscript for publication, provided you satisfactorily address the following data and other policy-related requests.

IMPORTANT - Please attend to the following:

a) Please change your title to "Eggs of the mosquito Aedes aegypti survive desiccation by rewiring their polyamine and lipid metabolism"

b) We note that you currently declare that you "received no specific funding for this work.” Please could you confirm whether this is correct, or supply appropriate funding details?

c) Please address my Data Policy requests below; specifically, we need you to supply the numerical values underlying Figs 1BCDE, 2BCDE, 3AB, 4ABCD, S1D, S2A, S3ABCD, S4BC, either as a supplementary data file or as a permanent DOI’d deposition. I note that you already have some data in the supplementary Tables, but their relationship to the individual Figure panels is unclear. Please could you clarify and/or supply the data required?

d) Please cite the location of the data clearly in all relevant main and supplementary Figure legends, e.g. “The data underlying this Figure can be found in S1 Data” or “The data underlying this Figure can be found in https://doi.org/10.5281/zenodo.XXXXX”

e) Please make any custom code available, either as a supplementary file or as part of a DOI'd deposition.

We expect to receive your revised manuscript within two weeks. 

*Published Peer Review History*

*Press*

Sincerely,

Roli Roberts

Roland Roberts, PhD

Senior Editor,

rroberts@plos.org,

PLOS Biology

DATA POLICY:

Regardless of the method selected, please ensure that you provide the individual numerical values that underlie the summary data displayed in the following figure panels as they are essential for readers to assess your analysis and to reproduce it: Figs 1BCDE, 2BCDE, 3AB, 4ABCD, S1D, S2A, S3ABCD, S4BC. NOTE: the numerical data provided should include all replicates AND the way in which the plotted mean and errors were derived (it should not present only the mean/average values).

CODE POLICY

Per journal policy, as the code that you have generated is important to support the conclusions of your manuscript, we require that you make it available without restrictions upon publication. Please ensure that the code is sufficiently well documented and reusable, and that your Data Statement in the Editorial Manager submission system accurately describes where your code can be found.

We require the original, uncropped and minimally adjusted images supporting all blot and gel results reported in an article's figures or Supporting Information files. We will require these files before a manuscript can be accepted so please prepare and upload them now. Please carefully read our guidelines for how to prepare and upload this data: https://journals.plos.org/plosbiology/s/figures#loc-blot-and-gel-reporting-requirements

DATA NOT SHOWN?

---

## [Editor Report · Decision Letter 3]

20 Sep 2023

Dear Dr Laxman,

Thank you for the submission of your revised Discovery Report "Eggs of the mosquito Aedes aegypti survive desiccation by rewiring their polyamine and lipid metabolism" for publication in PLOS Biology. On behalf of my colleagues and the Academic Editor, Mariana Wolfner, I'm pleased to say that we can in principle accept your manuscript for publication, provided you address any remaining formatting and reporting issues. These will be detailed in an email you should receive within 2-3 business days from our colleagues in the journal operations team; no action is required from you until then. Please note that we will not be able to formally accept your manuscript and schedule it for publication until you have completed any requested changes.

Sincerely, 

Roli Roberts

Senior Editor

PLOS Biology

rroberts@plos.org